# N2A Titin: Signaling Hub and Mechanical Switch in Skeletal Muscle

**DOI:** 10.3390/ijms21113974

**Published:** 2020-06-01

**Authors:** Kiisa Nishikawa, Stan L. Lindstedt, Anthony Hessel, Dhruv Mishra

**Affiliations:** 1Department of Biological Sciences, Northern Arizona University, Flagstaff, AZ 86011-5640, USA; Stan.Lindstedt@nau.edu (S.L.L.); dm2467@nau.edu (D.M.); 2Institute of Physiology II, University of Münster, 48149 Münster, Germany; Anthony.Hessel@uni-muenster.de

**Keywords:** mechano-sensing, muscle hypertrophy, muscle mechanics, muscular dystrophy, striated muscle, titinopathy

## Abstract

Since its belated discovery, our understanding of the giant protein titin has grown exponentially from its humble beginning as a sarcomeric scaffold to recent recognition of its critical mechanical and signaling functions in active muscle. One uniquely useful model to unravel titin’s functions, muscular dystrophy with myositis (mdm), arose spontaneously in mice as a transposon-like LINE repeat insertion that results in a small deletion in the N2A region of titin. This small deletion profoundly affects hypertrophic signaling and muscle mechanics, thereby providing insights into the function of this specific region and the consequences of its dysfunction. The impact of this mutation is profound, affecting diverse aspects of the phenotype including muscle mechanics, developmental hypertrophy, and thermoregulation. In this review, we explore accumulating evidence that points to the N2A region of titin as a dynamic “switch” that is critical for both mechanical and signaling functions in skeletal muscle. Calcium-dependent binding of N2A titin to actin filaments triggers a cascade of changes in titin that affect mechanical properties such as elastic energy storage and return, as well as hypertrophic signaling. The mdm phenotype also points to the existence of as yet unidentified signaling pathways for muscle hypertrophy and thermoregulation, likely involving titin’s PEVK region as well as the N2A signalosome.

## 1. Introduction: A Small Change in A Giant Protein Leads to Severe Titinopathy in mdm

Titinopathies [1] are inherited diseases of skeletal and cardiac muscle that are caused by mutations in the *Ttn* gene, which encodes for titin, the largest known protein [2]. Titin is the third most abundant protein in the muscles of vertebrates [3], and spans an entire half-sarcomere (1 μm) from the M-line to the Z-disk [4]. Titin plays many important roles in striated muscle, including passive force generation [5], maintenance of sarcomere integrity [6], and myofibrillar assembly [4,7]. Due to its large size and repetitive sequence, the *Ttn* gene exhibits enormous variability among humans [8,9]. Millions of potential isoforms are possible due to alternative splicing of the many (>360) *Ttn* exons [10]. Most of the disease-associated *Ttn* variants include mutations with large effects on the expressed titin protein, including nonsense, missense, and truncating mutations, insertions/deletions, and splice mutations [11]. In affected individuals, compound heterozygosity is common [12]. Despite the relatively large effects of these mutations on the expressed titin protein, many titin mutations are associated with relatively late onset myopathy and/or cardiomyopathy [11]. The diverse mechanisms of post-transcriptional and post-translational modification, and the diversity of signaling functions already described for this giant protein are staggering in number and complexity [13], which may help to explain why the underlying mechanisms through which titin mutations produce muscle disease remain largely unknown [14].

In contrast to more typical titinopathies, muscular dystrophy with myositis (mdm) in mice [15,16], among the earliest identified titinopathies [1,17], paradoxically presents a severe phenotype that is caused by a small deletion. Just 83 amino acids are missing from the giant titin protein [15], the largest isoform of which contains 38,000 amino acids. This represents a miniscule fraction (0.2%) of the entire protein. The mdm deletion is located at the N2A-PEVK border of I-band titin (Figure 1A). The N2A region of titin (Figure 1B) is comprised of four Ig domains and a unique insertion sequence (UN2A) in the order Ig80-UN2A-Ig81-Ig82-Ig83 [18]. In mdm, 21 amino acids are deleted from Ig83, and the remaining 61 amino acids are deleted from linking and PEVK regions (Figure 1B). Given the small size of the deletion, mdm is a surprisingly severe titinopathy with early onset shortly after birth and progressive degeneration, leading to early death [19]. Although the primary deletion is small, it remains to be determined whether splicing of the *Ttn* gene itself might also be affected. The severity of the phenotype suggests that this small region of titin plays a critical role in muscle function. A transgenic *TTN*^Δ112−158^ mouse was recently developed [20] in which 47 exons from the N-terminus of the PEVK region were deleted, including 28 amino acids at the C-terminus of the mdm deletion, yet life span and muscle function of these mice are normal [20]. Thus, it appears likely that only the remaining 55 amino acids deleted from Ig83 and the N2A-PEVK linker region are responsible for the severe mdm phenotype.

Lane [16] first reported the mdm mutation, which arose spontaneously on the C57BJ/6j mouse background at the Jackson Laboratories. Although the mdm mutation was initially mapped to chromosome 2, the affected gene(s) remained unknown [16]. When chromosome 2 was identified as the location of titin and nebulin genes [22], the hunt for the mdm mutation was quickly focused on these genes. Müller-Seitz et al. [22] collected titin and nebulin cDNA from mdm muscle and probed different regions for genetic mutations; however, no changes in titin or nebulin cDNA were uncovered. Nearly a decade later, with quickly advancing technology in sequencing, the site of the mutation was finally located within the titin gene [15].

Mdm is recessive lethal and first manifests in development as a kyphosis of the spine in homozygous mice at 12 days after birth [16]. Mdm mice exhibit a complex phenotype that, in addition to severe kyphosis, includes reduced body mass [19], rigid gait, and early death at approximately 60 days of age [15]. Histologically, signs of muscle degeneration appear in the soleus muscle within the first two weeks of birth, and 1–2 weeks later in the tibialis anterior [19]. However, regeneration of fibers is also evident from the presence of central nuclei in mdm muscles and the increased frequency and activity of satellite cells starting at 40 days of age, compared to wild type muscles [19].

The mdm mutation affects muscle function from myofibril to intact muscle [23]. Fiber bundles from mdm soleus have normal passive and active force at 24 days of age [24]. Single myofibrils have normal passive force, and only a small deficit in active isometric force at optimal length [25]. Fiber bundles have normal length- and calcium dependence of force [26], but also have increased passive tension and psoas fibers have reduced isometric tension compared to wild type muscles [27], although isometric tension is spared in extensor digitorum longus (EDL) fibers [26]. Intact muscles from mdm mice generally exhibit lower isometric tension, but the effect is greater in soleus than EDL [28].

In addition to changes in muscle structure and function, several recent studies demonstrate that the mdm mutation also affects post-natal growth and thermoregulation. While Witt et al. [24] assumed that mdm mice lose weight after birth, other studies suggest that they actually fail to grow after weaning. Huebsch et al. [29] found that skeletal muscle fibers from mdm mice remained small in diameter at post-natal day 14 when normal fibers have already undergone significant developmental hypertrophy. Pace et al. [30] also found that body mass failed to increase in mdm mice after weaning. Compared to wild type littermates, the growth of the long bones of the hindlimbs was also reduced after birth, although limb proportions remained relatively normal, indicating that growth of the skeleton is also reduced in mdm mice. These observations suggest impairment of normal hypertrophic signaling in skeletal muscles after birth, as well as concomitant changes in bone growth commonly observed in neuromuscular disease [31].

Two recent studies also demonstrate impaired thermoregulation in mdm mice. Taylor-Burt et al. [32] found that mdm mice failed to maintain a constant body temperature at ambient temperatures ranging from 20 to 37 °C, in contrast to wild type littermates. Homozygous mdm mice exhibited shivering thermogenesis at higher temperatures than wild type mice, but they shivered at a lower frequency than expected, particularly given their small body size. Miyano et al. [33] showed additionally that mdm mice have impaired non-shivering thermogenesis as well as reduced metabolic rate, indicating that the mdm deletion results in severe thermoregulatory defects at all levels including increased thermal conductance through decreased body size. Whereas earlier studies presumed that early death was due to failure of the respiratory muscles [15], the observation that provision of a heating pad and high fat diet increased the average lifespan from 60 to 120 days [32] suggests that thermoregulatory deficits may be the cause of early death. These recent studies demonstrate that titin mutations have significant organism-wide effects on phenotype beyond those traditionally considered in myopathy and cardiomyopathy.

Garvey et al. [15] were the first to suggest that the amino acid sequence deleted in mdm must play a critical role in titin function, perhaps as a mechanical hinge or linker, or as a binding site for ligands involved in signaling pathways. However, mechanical and/or biochemical mechanism(s) that account for the severe mdm phenotype have yet to be identified. Sufficient data are now available to critically analyze the accumulating evidence for biochemical signaling and mechanical functions of N2A titin that may account for mdm titinopathy, and also further our understanding of the role of titin in healthy muscle function.

## 2. Altered Gene Expression and Hypertrophic Signaling in mdm

Numerous studies have implicated N2A titin as a hub for signaling pathways [34,35,36]. If the mdm phenotype is caused by altered binding of ligands critical for signaling in healthy muscle, then we would expect to observe downregulation of proteins critical for healthy muscle function, and/or upregulation of pro-inflammatory, apoptotic, or other degenerative pathways. Downregulation of proteins critical for healthy muscle function appears to be minimal in mdm, with the only likely candidate being CAPN3. A transcriptomic analysis of 12,000 genes in hindlimb muscles from mdm mice showed that only four transcripts were downregulated [24]. Although CAPN3 transcripts were not downregulated in Witt et al.’s [24] study, two other studies revealed a deficiency in CAPN3 protein content within mdm muscles [15,37]. This deficiency is likely due to reduced binding of CAPN3 to mdm titin [24], since normal binding of CAPN3 to Ig83 in N2A titin suppresses CAPN3 autolysis [38].

Transcriptomic analysis of mdm muscles also fails to show upregulation of pro-inflammatory, apoptotic or other degenerative pathways that are evident in other muscular dystrophies. For example, in a mouse model of Duchenne muscular dystrophy (mdx), 72 different transcripts involved in the immune response are upregulated [39], whereas only three of these pro-inflammatory transcripts are upregulated in mdm [24]. Additionally, a utrophin/dystrophin double KO mouse has a degenerative phenotype that is more like the human Duchenne muscular dystrophy phenotype [40] than the mdx mutation in mice, but all three of the genes with the greatest upregulation in the utrophin/dystrophin double KO mouse are among the four downregulated genes in mdm.

Most studies that propose signaling-based mechanisms for the severe mdm phenotype have focused on CAPN3 and the MARP-family proteins due to their binding locations near the site of the N2A deletion and observed changes in gene expression. Surprisingly, Witt et al.’s [24] transcriptomic analysis also showed that MARP1/CARP and MARP2/Ankrd2, involved in hypertrophic signaling [41], are upregulated in mdm [24] despite the lack of developmental hypertrophy observed in mdm muscles [29,30]. Here, we also focus on these ligands that bind to titin in the N2A region near or within the *mdm* deletion [42], in particular calpain-3 CAPN3/p94 [43,44], and the MARP-family proteins MARP1/CARP, and MARP2/ankrd-2/Arpp [41]. Below, we review the observed changes in expression of CAPN3 and MARP-family transcripts or proteins in mdm muscles, with a view towards understanding the severe phenotype.

### 2.1. Calpain-3/p94 Deficiency in mdm

Calpain 3 (CAPN3, also called p94) is a calcium-dependent modulator protease [45] that is found throughout the body, but is expressed predominantly in skeletal muscle where it interacts with proteins throughout the sarcomere. CAPN3 binds to numerous muscle proteins in the Z-disk, thin filament (tropomyosin), and M-band, as well as at several locations along the length of titin [46]; for review see [45], including two locations at Ig80-IS and Ig82-Ig83 in the N2A region of titin [43,44]. Calpainopathies are muscular dystrophies that result from genetic mutations in CAPN3, revealing its critical function in healthy muscle [38]. In normal muscles, CAPN3 moves from the M-line to the N2A region of titin upon active stretch, and this movement is associated with exercise-induced muscle hypertrophy [47]. Substitution of a proteolytically inactive form of CAPN3 reduced the rate of translocation and impaired adaptation to exercise, demonstrating the important role of CAPN3 in strain- and exercise-dependent signaling in skeletal muscle.

Since mdm muscles exhibit CAPN3 deficiency [15,37], the mdm phenotype was initially thought to be linked to limb girdle muscular dystrophy type 2A, which is also characterized by CAPN3 deficiency often associated with CAPN3 null mutations [48]. Haravuori et al. [37] found that the concentration of CAPN3 protein was greatly reduced in mdm mice at 28 days of age, similar to muscle biopsies from tibial muscular dystrophy patients, whereas mdx muscles showed no reduction in CAPN3 expression. However, CAPN3 knockout mice have a much less severe dystrophic phenotype [49,50] than mdm mice, suggesting that functions in addition to CAPN3 are affected in mdm [15].

When mdm mice were crossed with overexpressing CAPN3 and CAPN3 transgenic knockout mice, Huebsch et al. [29] found that only the overexpressing CAPN3 genotype exacerbated the mdm phenotype, leading to a shorter life span and more severe muscular dystrophy. In contrast, the CAPN3 knockout/mdm double mutants showed no change in severity or progression of muscle degeneration, demonstrating that reduced CAPN3 expression is not a primary mechanism in mdm [29].

### 2.2. MARP-Family Proteins

MARP1 and MARP2 are key transcriptional regulators of hypertrophic signaling in striated muscle [51,52,53], and competitively bind to titin at UN2A-Ig81 [41]. In cardiac muscle, MARP2 plays a role in sequestering PKCα at the intercalated disc, and MARP2 knockout significantly worsens the dilated cardiomyopathy phenotype in a mouse MLP-KO model [54]. Since MARP 1 is typically found in trace amounts in healthy skeletal muscle, MARP2 is the predominant titin binding partner [55]. However, MARP1 is upregulated in response to a variety of mechanical stressors, including injury, stretch, denervation, and recovery from starvation [41,51,53]. When normal muscles experience a single damaging eccentric contraction, they upregulate five muscle-specific genes [51] including, MARP1, MARP2, and MLP. No upregulation of these transcripts is seen after isometric contraction or passive stretch. MARP1, MARP2, and MLP transcripts are upregulated in mdm, and MARP1 and MARP2 proteins are also upregulated [24].

Both hypertrophic signaling and MARP upregulation have also been associated with titin passive stiffness [53]. Van der Pijl et al. [53] investigated muscle hypertrophy in normal C57BL6j mice, and also in two transgenic models, including *RBM*20^ΔRRM^, which disables the splicing function of RBM20 and leads to a longer, more compliant titin isoform, and *Ttn*^ΔIAjxn^, which deletes 18 Ig domains at the AI junction, shortening the protein and creating a stiffer titin isoform. Unilateral diaphragm denervation was used to promote hypertrophy, in this case by passive stretching of the denervated hemi-diaphragm by the active side. The study demonstrated that muscles with the stiffer titin isoform showed both increased muscle hypertrophy and increased expression of MARP 1 and 2, versus muscles with the more compliant isoform, which showed attenuated hypertrophy and reduced expression of MARP 1 and 2, compared to wild type mice.

Since MARP1 and 2 are upregulated in mdm, it appears that some other factor associated with the mdm deletion prevents hypertrophy in mdm skeletal muscles, especially given that while the mdm mutation reduces binding of CAPN3, it has no effect on MARP binding [24]. Surprisingly, hypertrophic signaling pathways appear to function normally in transgenic mice in which all MARPs 1–3 have been deleted [56]. These mice show normal or even accelerated recovery from a single damaging contraction despite the total absence of MARPs. Likewise, *Ttn*^Δ112−158^ transgenic mice, with a shorter and stiffer PEVK region, add sarcomeres in series without upregulation of MARPS or other known hypertrophic signaling proteins [20]. These results suggest the existence of multiple parallel signaling pathways for hypertrophy in skeletal muscle, only some of which are known at the present time.

### 2.3. Mechano-Sensing in N2A Titin: Interactions between N2A Titin, MARPs, and CAPN3

Several recent studies have suggested that CAPN3 and MARPs form signalosomes on N2A titin that transduce mechanical stretch into nuclear gene expression signals related to muscle hypertrophy. MARPs, myopalladin, and CAPN3 colocalize in titin’s N2A region (Figure 2) and appear to be components of a mechano-sensing complex [41]. Miller et al. [41] speculate that MARPs are regulated by stretch of N2A titin, linking stress/strain signals in N2A titin to MARP-based regulation of gene expression. The close proximity of MARPs and CAPN3 suggests that titin could coordinate these signaling pathways. MARP1/CARP binds to the UN2A-Ig81 region of titin [57]. Using single molecule force spectroscopy, Lanzicher et al. [58] demonstrated that MARP1/CARP increases the persistence length and unfolding force of UN2A, suggesting that it might function as a chaperone that protects Ig81 from unfolding under high forces. In the diaphragm, van der Pijl et al. [53] proposed several possible functions of MARP1 including modulation of titin stiffness, cross-linking of titin filaments, or formation of a scaffold for other binding proteins to facilitate hypertrophic signaling.

The function of CAPN3 within muscle sarcomeres remains obscure [45]. Both N2A titin and MARPs are potential substrates for CAPN3 [43]. Due to the function of CAPN3 and other calpains as modulator proteases [45], it has been suggested that CAPN3 might cleave MARP1 and thereby increase its affinity for titin [53]. In cell culture, Hayashi et al. (2008) showed that the mdm deletion possessed enhanced resistance to proteases, including CAPN3, and they also reported that interaction between mdm titin and MARP1 was weakened.

In summary, despite the strong evidence for the N2A region of titin as a signalosome for mechano-sensing in skeletal muscle sarcomeres [53], whether the severe phenotype of mdm skeletal muscles is related to this function remains unclear for several reasons. The idea that calpain deficiency is a disease mechanism in mdm is not supported by crosses with transgenic CAPN3 over-expressing and KO mice, which demonstrated that the over-expressing mutation but not the KO background, exacerbated the mdm phenotype [29]. Furthermore, although mdm skeletal muscles exhibit significant upregulation of MARP1 and MARP2 [24], the normal developmental hypertrophy that occurs in wild type muscles, as well as in CAPN3 and MKO transgenic muscles, nevertheless fails to occur in mdm muscles [29,30]. In the context of a signaling function for N2A titin that is impaired by the mdm deletion, it appears likely that a mechanical signal is present that leads to an increase in MARP1 and MARP2 expression, but a process between these MARPs and other regulators of muscle hypertrophy may be impaired, which prevents the hypertrophy that is normally associated with MARP upregulation. It also remains possible that the primary mdm deletion in exons 106–109 may also affect splicing of titin exons in the PEVK region, which also binds to actin independently of calcium in the skeletal muscle [59].

## 3. N2A Titin: A Mechanical Switch in Skeletal Muscle

A second, non-exclusive alternative hypothesis for the severe mdm phenotype is that the N2A region of titin is a critical hinge or linker, so that the mdm deletion disrupts the mechanical function of titin in muscle sarcomeres in addition to hypertrophic signaling. Titin has traditionally been viewed as a spring in passive muscle [5,60,61]. However, a variety of studies demonstrate that titin stiffness increases in the presence of calcium [62,63,64]. Several recent studies further suggest that titin is a tunable spring that increases its stiffness substantially upon muscle activation [65]. It is now apparent that the N2A region of titin is involved in calcium-dependent binding with actin, functioning as a switch that modulates titin stiffness in passive vs. active muscle [66,67,68,69]. Here, we review the changes in passive and active muscle mechanics associated with the mdm deletion.

### 3.1. Titin: A Molecular Spring in the Resting Muscle

In skeletal muscle, titin’s I-band region is composed of two serially linked spring elements: tandem immunoglobulin (Ig) segments and the PEVK domain, separated by the N2A region. When a sarcomere is stretched, the I-band region of titin extends, giving rise to passive muscle force [60,62,70]. At relatively short sarcomere lengths (<2.8 μm), passive stretch straightens the proximal tandem Ig domains at low force [70,71]. Importantly, because proximal tandem Ig domains are so compliant, there is little to no change in passive tension with stretch. At longer sarcomere lengths (>2.8 μm), the much stiffer PEVK domain elongates and passive tension increases sharply [61,72]. In this way, the force-extension behavior of I-band titin determines the passive elasticity of skeletal muscle fibers over the physiological range of sarcomere lengths (2.4−3.2 μm) [70,73,74,75].

The titin isoform expressed by a muscle determines not only the sarcomere length at which passive tension develops [70,73] but also the magnitude of muscle passive stiffness. For example, soleus muscle fibers that express an isoform with a long, compliant PEVK region are less stiff than EDL or psoas fibers, which express an isoform with a shorter PEVK segment [76,77]. Transgenic mice with reduced (i.e., RBM20^ΔRRM^, [53]) or increased (i.e., *Ttn*^ΔIAjxn^, [53]; *Ttn*^Δ112−158^, [20]) titin stiffness also exhibit altered stiffness not only at the level of single muscle fibers but also at the level of the whole muscle.

Several studies demonstrate that the mdm mutation leads to fibrosis, or increased passive tension due to higher collagen content in the extracellular matrix. However, no difference in passive stiffness is observed between myofibrils from mdm and wild type mouse psoas muscles [25], consistent with the small size of the mdm deletion. However, increased passive tension is observed in preparations at all higher levels from single fibers to whole muscles, where endomysial, perimysial and/or epimysial connective tissue is present. For example, Lopez et al. [78] found that the passive tension of fiber bundles from the mdm diaphragm was elevated compared to wild type by 14 days of age. Furthermore, Powers et al. [27] found that passive tension nearly doubled and collagen content increased 10-fold in skinned fibers of mdm psoas compared to wild type. Passive tension is also higher in intact soleus [79,80] and EDL muscles [28] compared to wild type muscles.

### 3.2. Titin: A Tunable Spring in Active Muscle

Recent evidence demonstrates that the N2A region of titin is a mechanical switch that increases titin stiffness in an active muscle. A recombinantly expressed N2A construct (Ig81-UN2A-Ig81-Ig82-Ig83) was developed that co-sediments with F-actin [81]. Single molecule force spectroscopy demonstrates that rupture force increased from 70 pN at pCa = 10.0 to 100 pN at pCa = 4.0, and the off-rate of the N2A titin construct from actin decreased from 15.6 to 4.7 s^−1^ [81]. The high rupture forces (100 pN) measured between the N2A construct and F-actin in the presence of calcium show that titin alone can account for 66% of the energy stored in muscle during an active stretch that remains unexplained by other mechanisms including cross bridges [82]. The off-rate in the presence of calcium (4.7 s^−1^) is slow enough to explain most but not all of the long-lasting stress relaxation observed after a stretch of an active muscle [83].

These results demonstrate that N2A titin, at or around Ig83, binds to actin in the presence of calcium, and suggest a new mechanism that can explain the dynamic response of muscle to active length changes. In active skeletal muscle sarcomeres, N2A-actin binding increases titin stiffness and decreases titin equilibrium length by preventing low-force straightening of titin’s proximal Ig domains [84] so that only the stiffer PEVK region of titin extends during active stretch (Figure 3) [23,69]. Previous efforts to confirm that these interactions occur within muscle sarcomeres using fluorescently labeled N2A titin antibodies have been hampered by antibody cross-linking artifacts [68]. While direct observations of N2A titin await further technical developments, a variety of evidence from muscle mechanics strongly supports the existence of calcium-dependent titin-actin interactions in active muscle [81].

Evidence for an activation-dependent increase in titin stiffness is found both at the onset of muscle activation and following muscle deactivation. Muscle stiffness increases in the early stages of activation during the latent period between calcium release and cross-bridge attachment [63,85,86,87,88], and this early increase in stiffness is greater in muscles that express stiffer titin isoforms [89]. Enhanced tension persists following deactivation of previously stretched muscle fibers and intact muscles, so called “passive force enhancement” [90,91]. That this passive force enhancement is attributable to titin is supported by the observations that it persists after troponin C extraction, which prevents cross-bridge formation, and is abolished by mild trypsin, which selectively digests titin [92]. A large and growing body of evidence also demonstrates that titin contributes to the extra force produced by muscles following active stretch [93]. For example, force enhancement after active stretch of myofibrils increases with the stiffness of the titin isoform that they express [94].

### 3.3. Altered Mechanics of mdm Muscles

The mdm deletion, which includes Ig83 in N2A titin, is predicted to disrupt the titin-actin binding that normally occurs during muscle activation. When N2A titin binds to thin filaments upon calcium influx in muscle sarcomeres (Figure 3), two mechanical events occur. First, the length of titin’s freely extensible I-band must decrease upon activation [95,96] because the bound N2A allows only the PEVK region to extend during an active stretch (Figure 3, right). Second, because titin stiffness in passive muscle is limited by low-force straightening of the proximal tandem Ig domains [84], binding between thin filaments and N2A titin will increase titin stiffness in active muscles, as only the relatively stiffer PEVK region extends during stretch (Figure 3, [96]). Failure of proximal tandem Ig domains to straighten during active stretch will shift the force-extension curve leftward compared to the passive, low-calcium condition [21,79]. Indeed, there is experimental support for both of these mechanical events in wild type skeletal muscles and, critically, both fail to occur in skeletal muscles of mdm mice with a deletion in Ig83 (see below).

#### 3.3.1. Muscle Equilibrium Length Decreases upon Activation, but Not in mdm

Observations on elastic recoil of active muscle during rapid unloading are consistent with predictions from Ca^2+^-dependent N2A—actin binding that the equilibrium length of I-band titin should decrease when muscles are activated. When frog muscles activated at optimal length are rapidly unloaded, they recoil elastically by up to 20% of their length, ~77 nm per half sarcomere [97]. Similarly, single fibers from the telson muscle of horseshoe crabs recoil by only 6% of their length during rapid unloading, however they shorten by 210 nm per half-sarcomere due to the extremely long length (7 μm) of the sarcomeres in these muscles [98]. The large stretch of elastic elements in these muscles far exceeds any plausible limit in the cross bridges or filament lattice [97]. Monroy et al. [79] found that equilibrium length after elastic recoil was 15% shorter in active than in passive mouse soleus muscles, and there was a 2.9-fold increase in the slope of the stress–strain relationship during unloading. In contrast, soleus muscles from mdm mice showed no change in equilibrium length and no effect of activation on the slope or intercept of the stress–strain relationship during unloading. These results are consistent with Ca^2+^-dependent binding of titin to actin reducing the titin free length and increasing titin stiffness, but not in the mdm muscles.

#### 3.3.2. Titin Force and Stiffness Increase upon Activation, but Not in mdm

Titin stiffness increases in active compared to passive myofibrils. Leonard and Herzog [95] examined the force-extension behavior of single myofibrils stretched beyond the overlap of the thick and thin filaments, in which no contribution of cross bridges is possible. In these experiments, single myofibrils with four to twelve sarcomeres in series were mounted between a needle for controlled length changes and a cantilever to measure force. Myofibril lengths were adjusted to an initial average sarcomere length of 2.5 μm. Individual sarcomere lengths were measured at the end of the stretch to confirm the loss of filament overlap. They found that the slope of the force–extension relationship was greater in calcium-activated myofibrils than in passively stretched myofibrils. Subsequent studies [99] confirmed the basic result in mouse psoas. 

The question arises whether collagen or intermediate filaments (e.g., desmin, etc.) could contribute to force of myofibrils stretched to long sarcomere lengths. In general, intermediate filaments are found at the level of single fibers, in which they bear considerable passive force at sarcomere lengths > 2.8 μm [77]. However, at the level of a single myofibril, it is difficult to see how intermediate filaments could contribute to passive or active force. They are not in series with actin and myosin, nor do they extend as a continuous filament along the length of a single myofibril, as does titin. In addition, short-duration, mild trypsin digestion has been shown to selectively degrade the extremely thin (4 nm diameter) and extended (1.2 μm length) titin filament, and completely abolishes force in passive and active myofibrils stretched beyond overlap of thick and thin filaments [25,95,99].

Significantly, the mdm mutation prevents the increase in titin-based stiffness (Figure 4, [25]) that normally occurs upon activation of wild type myofibrils [95,99].

#### 3.3.3. Muscle Force Increases with Active Stretch and Decreases with Active Shortening, but Not in mdm

Ca^2+^-dependent titin-actin interactions provide a relatively simple explanation for history-dependent muscle properties, including residual force enhancement [21,95] and force depression [23,96,100]. Indeed, both force enhancement and force depression are negligible in skeletal muscles from mdm mice compared to passive tension at the final length (Figure 5, [80]). The mdm mutation has also been shown to reduce negative work in eccentric stretch–shortening cycles [66] and positive work in concentric stretch shortening cycles [69]. In contrast, length dependence of calcium activation (Figure 6, [26]) and the force-velocity relationship [80] are unaffected by the mdm mutation. The velocity of shortening under constant load is non-linear and decreases significantly with length, especially at the smallest loads and highest velocities, supporting the existence of an opposing force due to N2A – actin binding (see [23]; Fig. 1B). Therefore, mdm muscles would be expected to have an impaired force-velocity relationship. However, the state of the art in the field is to estimate the velocity at a given load from the initial linear velocity near L_0_, which likely explains the similarity between WT and mdm muscles in the force-velocity relationship.

### 3.4. Conclusion: Mechano-Signaling Functions of N2A Titin and Mechanisms of Titinopathy

Since the discovery that the mdm phenotype results from a small deletion in the N2A region of titin [15], it is increasingly apparent that the affected region plays a critical role in skeletal muscle function. Garvey et al. [15] suggested that this 83 amino acid deletion at the border between N2A and PEVK regions of titin might serve as a site for binding of crucial ligands or as a critical mechanical linker or hinge. Over the ensuing decades since 2002, evidence has accumulated that the N2A region is an important hub for “mechano-sensing,” transducing titin stress or strain into biochemical signals that alter gene expression [41,53]. Currently, substantial evidence supports both mechanical and signaling functions, two non-exclusive hypotheses for critical roles of N2A titin.

#### 3.4.1. Effects of mdm on Hypertrophic Signaling

The idea that N2A titin is a hub for hypertrophic signaling in skeletal muscles is based on colocalization of the modulatory protease CAPN3 with hypertrophic signaling proteins including MARPs 1 and 2, and myopalladin [41,53,58]. The strongest evidence in support of the existence of an ‘N2A signalosome’ comes from unilateral diaphragm denervation (UDD), in which a passive stretch of the denervated hemi-diaphragm by the active side leads to hypertrophy of the denervated side [53]. When UDD was investigated in three mouse genotypes, the amount of hypertrophy increased with both titin stiffness and upregulation of MARP1 and MARP2.

The mdm mutation appears inconsistent with this model of hypertrophic signaling since both MARP 1 and 2 are upregulated, but muscle hypertrophy is paradoxically absent. However, a recent study on myopalladin knock-out mice provides a possible resolution [101]. Like mdm mice, myopalladin KO mice are smaller than their wild type littermates and their muscle fibers are also smaller in diameter, although their body mass is only 17% smaller than the wild type at 28 days and the difference in weight decreases with age. In contrast, mdm mice have a 50% smaller body mass than wild type at 24 days [24] and the difference increases with age [30]. Nevertheless, a scenario is possible in which the mdm deletion prevents CAPN3 binding to Ig83 in N2A titin. Subsequently, loss of CAPN3 binding could affect the MARP structure and lead to dissociation of myopalladin from the signalosome (see Figure 2), leading to a loss of hypertrophy in mdm as observed in myopalladin KO.

Nevertheless, disruption of a possible CAPN3-MARP-MYPD signalosome in N2A titin by the mdm mutation cannot alone account for the N2A phenotype for several reasons. First, the mdm phenotype is more severe than the phenotype of CAPN3 knockout [29], MARP triple knockout [56], or myopalladin KO mice [101] Secondly, none of these genes have upregulated expression in *Ttn*^Δ112−158^, which nevertheless exhibits hypertrophic addition of sarcomeres in series [20]. Furthermore, the defects in shivering thermogenesis, non-shivering thermogenesis, homeothermy, and basal metabolic rate associated with mdm suggest a potential novel role for titin in regulation of muscle-based thermogenesis [33]. In summary, the available evidence suggests that while mdm likely affects the function of the N2A ‘signalosome’, other signaling pathways associated with muscle hypertrophy and thermoregulation must also be also affected.

#### 3.4.2. Effects of mdm on Muscle Mechanics

In order for titin to contribute to muscle passive tension and maintain sarcomere integrity at physiologically relevant sarcomere lengths, binding of N2A titin to actin is necessary to prevent elongation of the proximal tandem Ig domains at very low force [65]. Consistent with this requirement, recent studies demonstrate that the binding affinity and off rate of N2A titin–actin interactions increases in the presence of Ca^2+^ (Dutta et al., 2018) and that this interaction likely involves Ig83, which is partly deleted in mdm. Likewise, the mechanics of mdm muscles are consistent with predicted changes in muscle equilibrium length and stiffness upon activation [23,69]. These deficits, predictably, prevent the increase in titin stiffness in contracting myofibrils stretched beyond overlap of thick and thin filaments [25], the change in equilibrium length of active muscles during rapid unloading [79], force enhancement following active stretch, force depression following active shortening [80], and energy dissipation (negative work) during stretch–shortening cycles [66] that normally occurs in wild type muscles.

Calcium-dependent interactions between N2A titin and actin must have profound effects on hypertrophic signaling and the N2A titin signalosome, since the UN2A region (see Figure 3) should be protected from a stretch in the active muscle, whereas hypertrophy due to stretching of the passive muscle would be unaffected. If only the PEVK region of titin extends during a stretch of the active muscle, then the PEVK region, rather than N2A, should function as a mechano-sensor. In contrast to the N2A region, where several signaling proteins colocalize, potential binding partners for the PEVK region are more limited. However, several studies demonstrated that PEVK binds to the SH3 domain of nebulin [102,103], suggesting that signaling proteins with SH3 binding domains might interact with PEVK titin [104].

In summary, the 21st century has seen remarkable progress in understanding of the signaling and mechanical functions of N2A titin. However, known mechanisms fall far short of fully accounting for the severe mdm phenotype. The severity of functional disruption caused by this small deletion in the giant titin protein demonstrates that the current paradigm is incomplete. The idea that N2A titin normally functions as a mechanical switch upon muscle activation, but not in mdm, leads toward new directions in the search for mechanosensing functions of titin in muscle health and disease.

## Figures and Tables

**Figure 1 ijms-21-03974-f001:**
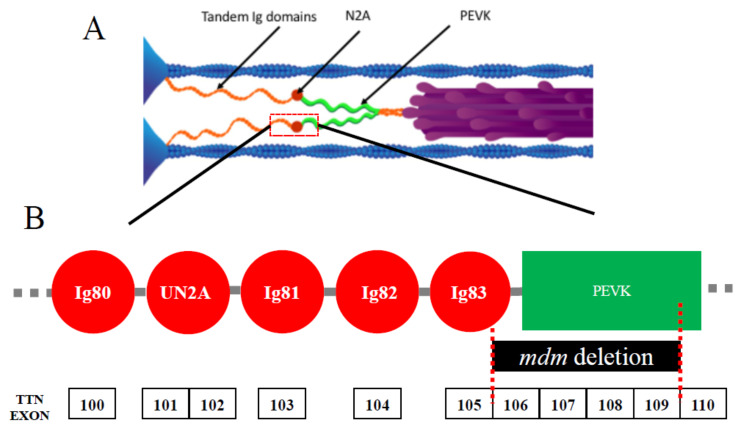
(**A**) Layout of titin in skeletal muscle sarcomeres. Each titin molecule is bound to the thin filament (blue) in the Z-disk (blue triangles on left side of half-sarcomere) and to the thick filaments (purple) in the A-band. The N2A segment (red) is located between the proximal tandem Ig segments (orange) and the PEVK segment (green). Ig, immunoglobulin. Reproduced with permission from Nishikawa [21]. Copyright 2016, The Company of Biologists Ltd. (**B**) Layout of the N2A region. This region includes Ig80, unique N2A sequence (UN2A), Ig81, Ig82, and Ig83. The exons of the mouse *Ttn* gene are shown below, along with the location of the mdm deletion.

**Figure 2 ijms-21-03974-f002:**
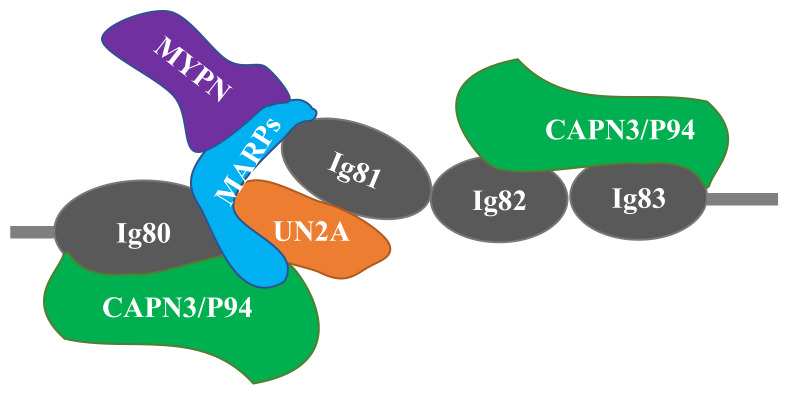
Schematic of the titin-N2A signaling complex. MARP1 and 2 bind competitively to N2A titin’s unique sequence (UNA), located between Ig80 and Ig81. CAPN3 binding is localized to both Ig80/UN2A and Ig82/Ig83. Myopalladin (MYPN) associates with the N-terminal domains of MARPs. Adapted from Miller et al. [41] and Hayashi et al. [43].

**Figure 3 ijms-21-03974-f003:**
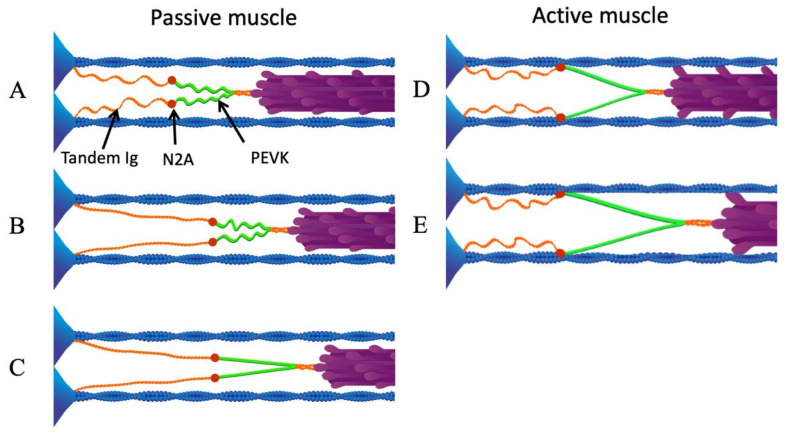
N2A—actin interaction eliminates low-force straightening of tandem Ig domains. (**A**) Passive sarcomere at slack length. (**B**) As the sarcomere is stretched passively beyond its slack length, the proximal tandem Ig segments unfold approximately to their contour length. (**C**) After the proximal tandem Ig segments have reached their contour length, further passive stretching extends the PEVK segment. (**D**) Upon activation, N2A titin binds to actin. (**E**) When the active sarcomere is stretched, only the PEVK segment (green) extends. Reproduced with permission from Nishikawa [21]. Copyright 2016, The Company of Biologists Ltd.

**Figure 4 ijms-21-03974-f004:**
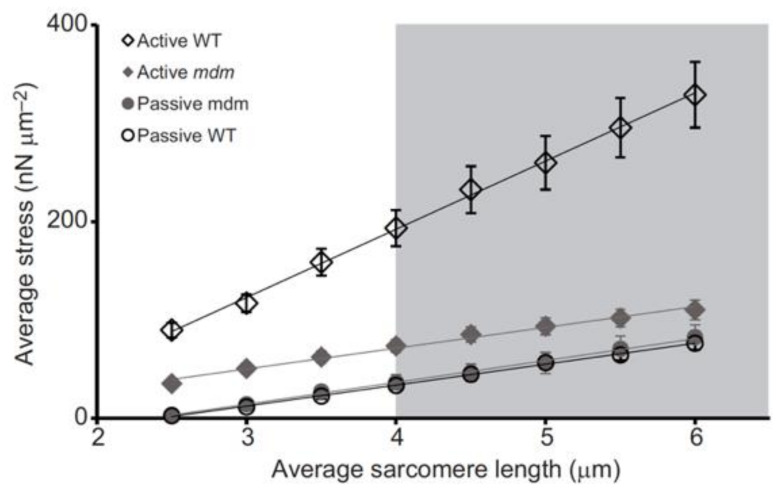
Activation increases titin-based force and stiffness of wild type psoas myofibrils, but not mdm myofibrils, during stretch. Stress (nN/μm^2^) vs. sarcomere length (μm) of single psoas myofibrils from the wild type (WT, black open symbols) and muscular dystrophy with myositis (mdm, gray filled symbols) mice during active (diamonds) and passive stretch (circles). There is no difference in passive stress between WT and mdm myofibrils. Actively stretched WT myofibrils are stiffer than active mdm and both passive mdm and WT, while actively stretched mdm myofibrils do not differ in stiffness from passively stretched mdm and WT myofibrils. Reproduced with permission from Powers et al. [25]. Copyright 2016, The Company of Biologists Ltd.

**Figure 5 ijms-21-03974-f005:**
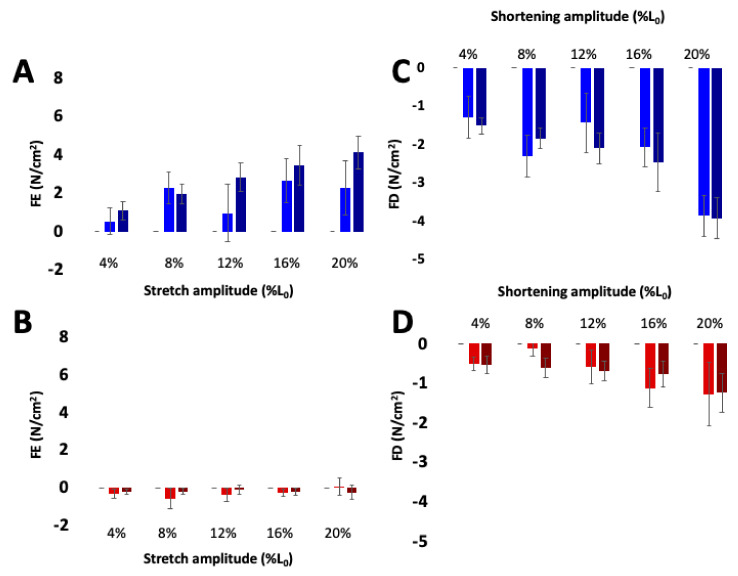
Residual force enhancement (**A**,**B**) and depression (**C**,**D**) increase with activation in wild type (blue) but not mdm (*red*) soleus muscles. The contribution of passive stress was subtracted from residual force enhancement and depression. Darker bars indicate maximal activation and lighter bars indicate submaximal activation. Reproduced with permission from Tahir et al. [80]. Copyright 2019, The Company of Biologists Ltd.

**Figure 6 ijms-21-03974-f006:**
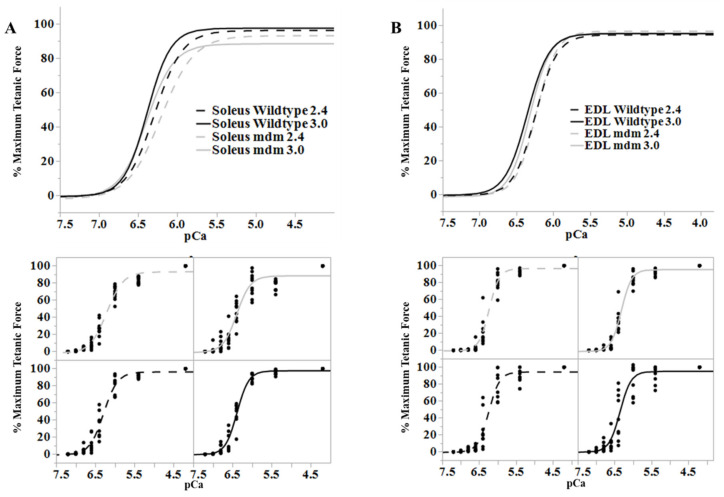
Length-dependence of Ca^2+^-sensitivity is similar in the wild type (black) and mdm (gray) fiber bundles. (**A**) Soleus and (**B**) extensor digitorum longus (EDL). Ca^2+^ sensitivity relationships at sarcomere lengths of 2.4 (dashed lines) and 3.0 µm (solid lines) from wild type and mdm fiber bundles (top). Individual data points for each curve are shown below. The shift in Ca^2+^-sensitivity from 2.4 to 3.0 µm was similar between genotypes. Reproduced with permission from Hessel et al. [26]. Copyright 2019, The Company of Biologists Ltd.

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
