# Peer review of "N2A Titin: Signaling Hub and Mechanical Switch in Skeletal Muscle"

_ijms, 2020, doi:10.3390/ijms21113974_

Round 1
Reviewer 1 Report
This is a well-done and timely review and I have relatively minor comments.
- There is an extensive review of results of the mdm mouse model , with multiple phenotypic changes discussed. Some of these could very well be secondary changes (like the altered thermoregulation) and not due to titin per se. Please discuss this topic, separating changes that are seen in young mice and therefore more likely primary changes, from those that are seen in older mice and more likely representing secondary changes.
- Explain better how altered calcium-dependent binding of N2A titin to actin filaments can underlie the mdm mouse phenotype.
- In the discussion of the role of calpain it is also worth discussing the distribution of calpain (and not just its abundance) please discuss the work by the late Sorimachi and colleagues in PMID: 20592470.
- Does direct evidence for binding of the N2A region to the thin filament in an activated sarcomere (immunolabeling) exist? If so, please discuss. If not, please discuss that up until obtaining direct evidence, the model in Figure 3 (active muscle) is speculative and requires direct testing.
- The direct effect of calcium on titin’s stiffness should be discussed in greater detail, including the work in PMID: 15548726 and PMID: 26405100.
- Line 373. It is unexpected that the force-velocity relation is not altered in mdm mice since one would expected that strong N2A-thin filament binding would impede shortening as it will function as an opposing force once the thick filament tip is proximal to the N2A-thin filament interaction point. Please discuss.
- Figure 4 suggest enormous static forces exerted on actin in myofibrils stretched far beyond overlap (60-100 pN per titin molecule). It is hard to believe that such static forces would be held by N2A-actin interaction. Has this result in Figure 4 been reproduced in other laboratories? Please describe this figure in greater detail, including how these experiments were actually performed. Forces far beyond overlap could also be generated by intermediate filaments and are not necessarily all titin-based. Please discuss.
- Minor: line 43 misspelling ( describednfor). Line 55, isn’t’ technically the first titinopathy identified by Gerull et al (PMID: 11788824)? Line 85, also discuss Witt et al. who also measured passive and active force in mdm mice, line 185 and elsewhere the ‘deltas’ do not shown, line 253 and beyond, PMID: 9472037 also should be cited. Line 375, remove ‘from’,
Reviewer 2 Report
General comments.
This review brings well together the tremendous progress on the titin N2A segment, coming from mouse molecular genetic studies, protein biochemistry, cell biology, and myofibril mechanics experimentation. This review is ideally suited for novel workers in the field that want to gain a detailed overview on this interdisciplinary research topic in basic myology. I warmly recommend this review for publication in IJMS. Before publication, the following specific points should be addressed.
Specific points.
P1, line 7, typo in münstser.
P1, line 14 animal model “.. by accident”. I think giving the specifics makes it clearer, i.e. by a transposon-like LINE repeat insertion.
Line 17, “all aspects”. I think “diverse aspects” is more appropriate.
Page 2, line 43, typo.
Page 2, Fig. 1B: Since Fig1 is a color figure anyway, I think it would make sense to color the IG, N2A-us, and PEVK domains in 1B accordingly as the colors for the titin regions given in 1A.
Page 2. Line 55, “the earliest identified titinopathy”: I suggest to delete these four words. Alternatively, cite also Gerull et al. and Hackman et al for the human titinopathies that were described the same time and I believe were actually submitted earlier (Gerull B et al., Nature Genetics 2002; . 30: 201-204 for DCM; Hackman P et al. 2002 for TMD: Am J Hum Genet. 71: 492-500). Recommendation: Cite all three genetic titinopathies described in 2002.
Line 57, discussion of ref 17. Garvey at al show data that the 83-residue deletion is significantly affecting PEVK splicing patterns, thus leading to the exclusion of larger PEVK segments in most splice isoforms. As the titin splicing patterns are highly complicated, its analysis and discussion goes beyond this review. However, I recommend to state here that in addition to the 83-residue deletion, the splicing pattern of the PEVK region is affected, leading to much larger truncation events of the PEVK spring region. Consequently, I recommend to down-tune the argument on the deletion of a miniscule segment. At least mention the splice-stiffening hypothesis. The argument that the transgenic model in 20 has no severe phenotype is in my opinion not valid: This model was designed to create an in-frame deletion of PEVK exons, and the targeting cassette was taken out after gene targeting. Whereas the mdm mutation occurred by insertion of a large LINE repeat, most likely significantly affecting RNA processing, thereby perturbing correct splicing.
Line 131, ref 27, Witt et al. I think the main finding reported by Witt et al were that CARP was strikingly upregulated on the protein level on top of its targeting to the central I-band region/ titin N2A region (in contrast to the non-titin based Mdx model). I suggest to discuss this when referring to ref 27, as this aligns well to the titin N2A cartoon and the MARP targeting. It is not uncommon that expression levels as evaluated on the transcriptome level by arrays as by Witt et al and actual proteome levels analyzed later vary significantly.
Line 197, ref 53: It is indeed very surprising that deletion of all three MARP genes appears not to cause a significant phenotype. I wonder therefore if during the several years of producing all six MARP null alleles and then breeding them together a phenotypical rescue/ positive selection may have happened, possibly masking a phenotype. Please note that Ju Chen´s group subsequently reported that the deletion of CARP has a striking effect on the severity of the DCM phenotype in the MLP-KO model (Lange S, Chen J). You may cite this as evidence on the importance of the MARP member CARP in certain disease contexts.
Line 241: This second non-exclusive hypothesis also aligns well with the idea that the mdm deletion has secondary effects on PEVK splicing, thus stiffening this part of the titin spring. Titin-PEVK may as well interact with actin, as titin N2A. I suggest to add this hypothesis here, i.e. that the mdm deletion may affect the titin PEVK spring as well, before moving to the molecular spring review section.
